# Procyanidins and Anthocyanins in Young and Aged Prokupac Wines: Evaluation of Their Reactivity Toward Salivary Proteins

**DOI:** 10.3390/foods14101780

**Published:** 2025-05-17

**Authors:** Katarina Delić, Danijel D. Milinčić, Aleksandar V. Petrović, Slađana P. Stanojević, Anne-Laure Gancel, Michael Jourdes, Mirjana B. Pešić, Pierre-Louis Teissedre

**Affiliations:** 1Institute of Food Technology and Biochemistry, Faculty of Agriculture, University of Belgrade, Nemanjina 6, 11080 Belgrade, Serbia; katarina.delic@u-bordeaux.fr (K.D.); danijel.milincic@agrif.bg.ac.rs (D.D.M.); sladjas@agrif.bg.ac.rs (S.P.S.); 2Bordeaux Sciences Agro, Bordeaux INP, Université de Bordeaux, UMR 1366 OENOLOGIE, ISVV, 33140 Villenave d’Ornon, France; anne-laure.gancel@u-bordeaux.fr (A.-L.G.); michael.jourdes@u-bordeaux.fr (M.J.)

**Keywords:** young wine, aged wine, Prokupac, salivary proteins, procyanidins, anthocyanins

## Abstract

In this study, the reactivity of procyanidins and anthocyanins in young and aged Prokupac wines toward salivary proteins is investigated via SDS-PAGE and UHPLC-QTOF-MS to determine the differences between the phenolic compounds of red wine in relation to the aging process of wine. SDS-PAGE analysis revealed that procyanidins, flavanol-anthocyanin polymers, and ellagitannins in aged wine have strong affinities for salivary proteins, leading to the formation of insoluble complexes. By contrast, young wine contained predominantly procyanidins with high salivary protein affinity, as well as monomeric flavan-3-ols and anthocyanins, which mainly form soluble aggregates, while polymeric phenolics were less represented. Electrophoretic patterns further showed that seed-derived procyanidins mainly formed insoluble complexes with salivary proteins, whereas skin-derived anthocyanins tended to form soluble ones. The total content of all phenolic compounds quantified by UHPLC-QTOF-MS was 2.5 times higher in young wine than in aged wine, primarily due to the significantly greater abundance of malvidine-3-*O*-glucoside in young wine (eightfold higher level in young wine). Targeted UHPLC-QTOF-MS analysis of selected phenolics confirmed the electrophoretic results and showed a higher binding affinity of procyanidins in aged wine compared to young wine, as well as a higher percentage of procyanidin binding compared to anthocyanins, independent of the age of the wine. Sensory evaluation showed that aged wine had higher tannin quality scores, whereas young wine exhibited greater acidity and astringency, with bitterness being comparable between them. These results highlight the influence of wine aging on the interaction between phenolic compounds and salivary proteins and emphasize the dominant role of procyanidins in protein binding and the potential synergistic contribution of anthocyanins to mouthfeel perception.

## 1. Introduction

Grapevine production and winemaking are among the largest agricultural and food technology industries worldwide. Global wine production for 2024 was estimated to be between 227 mhl and 235 mhl, while global wine consumption in 2023 was estimated at 221 mhl, making wine one of the most economically significant products for many countries [1,2]. Grapes and wine are rich sources of various phenolic compounds, which significantly contribute to their health benefits [3,4], quality, and sensory acceptance [5,6,7]. During red winemaking, phenolic compounds are easily transferred from grape juice to wine, enhancing the complexity of red wine, as well as its vibrant color and astringent mouthfeel.

Astringency is often an important sensory attribute of red wine, characterized by a typical dryness and puckering oral sensation. Previous research has shown that astringency results from the interaction between flavan-3-ols and their polymerized forms (proanthocyanidins) with salivary proteins, leading to the formation of soluble or insoluble complexes [6,8,9]. The widely accepted mechanism of astringency is based on the precipitation of salivary proteins by phenolic compounds [10]. Saliva is a complex and slightly acidic mucoserous exocrine secretion that consists of over 99% water and includes electrolytes, proteins, and nitrogenous products [11,12]. Salivary proteins, such as mucins, enzymes, immunoglobulins, proline-rich proteins, cystatins, histatins, statherins, and cathelicidins, play important multifunctional roles. Among them, proline-rich proteins, which constitute up to 70% of the proteins in human saliva, are crucial for interactions with phenolics. These proteins are highly polymorphic and are categorized into basic (bPRPs), acidic (aPRPs), and glycosylated (gPRPs) forms, differing in structure, size, and their tendency toward phenolic compounds and tannins [13,14,15,16]. For evaluation of phenolic reactivity toward salivary proteins/PRPs, most studies have used grape seed extract as a model system due to its high diversity of procyanidins [8,17,18,19,20,21,22]. These studies have shown that the chemical binding affinity of phenolics and proteins often strongly depends on the concentration of phenolics/proteins, as well as the degree of polymerization/galloylation of the monomeric flavan-3-ols [8]. Apart from pro(antho)cyanidins and tannins, recent studies have shown that other grape phenolic compounds can also interact with salivary proteins and may contribute to overall sensory in-mouth sensations [7,23,24]. For example, anthocyanins are the dominant phenolics in red wine and grape skins [4,25], so it is expected that they come into contact and interact with salivary proteins after ingestion. Several studies have demonstrated interactions between various grape skin anthocyanins and salivary proteins [24,25,26,27], leading to the formation of new soluble complexes that affect mouthfeel sensations. Regarding anthocyanin derivatives, acetylated, coumaroylated and cinnamoylated anthocyanins have been shown to increase astringency [24], while soluble complexes of some anthocyanin glucosides activate receptors for bitterness [28]. To date, only a few studies have used wine to investigate the mechanism of salivary protein interaction/precipitation with phenolic compounds [10,27,29], primarily focusing on wine procyanidins [30,31] in the presence or absence of polysaccharides [32]. However, red wine is a rich source of anthocyanins, flavan-3-ols, and procyanidins, which can have a synergistic effect on the interaction with PRPs, as previously reported by Soares et al. [33]. These compounds (procyanidins and anthocyanins) are also involved in co-pigmentation, polymerization, and condensation processes during wine aging and storage, which increase the content of pyranoanthocyanins and cause the color to shift from bright red hues to more subdue and brick-red tones [34,35,36]. The balance between procyanidins and proanthocyanidins evolves over time, and their interactions with anthocyanins and the formation of pyrano(antho)cyanins enhance the color stability and sensory profile of aged red wine [24,37]. In addition, ellagitannins extracted from oak wood during wine aging have shown a strong tendency to bind to salivary proteins, directly affecting the sensory characteristics of wine [38]. To better understand the combined effect and contribution of anthocyanins, flavan-3-ols, and procyanidins on the astringency of young and aging wines, further studies are necessary.

Bearing in mind previous observations, the aim of this study was to examine the chemical affinity and binding abilities of anthocyanins and procyanidins toward salivary proteins, using real systems such as young and aged Prokupac red wines. In addition, this study promotes the traditional red grape variety Prokupac grown in Serbia and wine from this variety, in accordance with the Agenda for Sustainable Developments, which involves the preservation of biodiversity. Previous research has shown that Prokupac seeds are a rich source of flavan-3-ols and different proanthocyanidins [4,39]. Wine from this variety displays unique sensory properties and aging potential, as well as typical astringent sensations that are primarily contributed by tannins derived from the seeds during fermentation. For these reasons, Prokupac wine can serve a good model for examining phenolics/salivary protein interactions. To determine the affinity of wine phenolics for salivary proteins, this study includes ultra-high-performance liquid chromatography coupled with triple/time-of-flight mass spectrometry (UHPLC-QTOF-MS) and sodium dodecyl sulfate polyacrylamide gel electrophoresis (SDS-PAGE).

## 2. Materials and Methods

### 2.1. Wine, Seed and Skin Samples

In this study, young and aged Prokupac wines were used to evaluate phenolics/salivary protein interactions. Wine from the 2024 vintage, collected immediately after fermentation and separation from pomace, was marked as young Prokupac wine (YPW). By contrast, bottled Prokupac wine from the 2021 vintage that had undergone two years of barrique maturation was labeled as aged Prokupac wine (APW). In addition, Prokupac grape seed and skin extracts were prepared and used as control samples, predominantly containing anthocyanins and procyanidins, respectively. These control samples were included to aid in the interpretation of the electrophoretic results obtained from the mixtures of wine and salivary proteins.

Preparation of the seed and skin extracts included the following steps: (1) peeling the grape skins and manually separating them from the seeds, followed by washing, freeze-drying, and fine grinding with a coffee grinder; (2) extracting skin (1 g) and seed (1 g) powders with 80% acidified methanol (1:10 *w*/*v*) for 1 h on a mechanical shaker, as previously reported by Milinčić et al. [4]. The supernatants obtained after centrifugation (4000× *g*, 10 min) were evaporated to dryness using a rotary evaporator under reduced pressure at 40 °C, dissolved in 10 mL of milliQ water, freeze-dried, and used for further analysis.

### 2.2. Saliva Collection

The saliva of 10 volunteers (5 men and 5 women, aged 24 to 47 years) was collected in the morning between 11 am to 12 noon, following the circadian rhythm. Saliva collection was performed in accordance with ethical permission using the method previously described by Paissoni et al. [24]. Briefly, the study and saliva collection were approved by the ethical committee of the Laboratory Research Unit USC 1366 of the Institute of Viticulture and Enology of the University of Bordeaux (ISVV). All participating volunteers signed a consent form with information about the type of the research, voluntary participation, and the spitting protocol. Participants were asked not to eat or drink for at least one hour before samples were taken. The saliva was then collected in Eppendorf tubes (15 mL), pooled, frozen at −20 °C, and freeze-dried.

Prior to the saliva test, freeze-dried saliva (10 mg/mL) was reconstituted in phosphate-buffered solution at pH 6.8, vortexed, and stored in a refrigerator at 4 °C for 1 h. The reconstituted solution was then centrifuged at 4000× *g* for 10 min to obtain the salivary protein solution (SP), which was used for mixing with wine and skin/seed extract samples.

### 2.3. Saliva Test

The binding test of wine phenolics with salivary proteins was performed according to the methodology previously described by Ma et al. [8] and Paissoni et al. [24], with a slight modification of the protocol. In brief, 4 mL of the two wine samples (YPW and APW) was mixed with 1 mL of salivary protein solution or 1 mL of phosphate-buffered solution (pH 6.8) and incubated at 37 °C for 5 min. After incubation, the mixtures were centrifuged at 17,000× *g* for 5 min. The collected supernatants were filtered through 0.22 µm nylon syringe filters and analyzed for selected anthocyanins and procyanidins via electrophoresis and targeted UHPLC-QTOF-MS. Control samples were taken after incubation of the SP/wine sample mixtures at 37 °C for 5 min and filtration through 0.22 µm nylon syringe filters.

The freeze-dried seed and skin extracts were prepared at a concentration of 1 mg/mL in the model wine solution (12% ethanol, 4 g/L tartaric acid, pH 3.5), mixed intensively for 1 h, and centrifuged at 3000× *g* for 5 min to remove any insoluble particles in the solution. Then, the skin and seed solutions (4 mL) were mixed with 1 mL of salivary protein solution and incubated at 37 °C for 5 min. After incubation, the mixtures were centrifuged at 17,000× *g* for 5 min. The collected supernatants were filtered through 0.22 µm nylon syringe filters and used for electrophoretic analysis. Control samples were taken after incubation of the SP and skin or seed sample mixtures at 37 °C for 5 min and filtration through 0.22 µm nylon syringe filters. Control salivary proteins were prepared by mixing 1 mL of salivary protein solution with 4 mL of model wine solution.

### 2.4. SDS-PAGE Analysis

In this study, sodium dodecyl sulphate-polyacrylamide gel electrophoresis under reducing conditions (SDS-R-PAGE) was performed to analyze the salivary proteins before and after interaction with wine, skin, and seed phenolics. For this analysis, separating gels (12.5% *w*/*v*; pH 8.85) and stacking gels (5% *w*/*v*; pH 6.8) as well as Tris-Glycine running buffer [0.05 M Tris, (pH 8.5), 0.19 M glycine, 0.1% *w*/*v* SDS] were prepared as previously described in detail by Pešić et al. [40]. The following salivary protein solution/wine samples were used for electrophoretic analysis:(a)Control salivary proteins—CSP;(b)Salivary protein solution/young wine filtrate (after filtration through 0.22 µm filter)—SP/YW-F;(c)Salivary protein solution/young wine precipitate (after centrifugation)—SP/YW-P;(d)Control young Prokupac wine—CYPW;(e)Salivary protein solution/aged wine filtrate (after filtration through 0.22 µm filter)—SP/AW-F;(f)Salivary protein solution/aged wine precipitate (after centrifugation)—SP/AW-P;(g)Control aged Prokupac wine—CAPW.

The following salivary protein solution/seed extracts and salivary protein solution/skin extracts were used for electrophoretic analysis:(a)Salivary protein solution/seed extract after incubation (37 °C, 5 min)—SP/SE-I;(b)Salivary protein solution/seed extract filtrate (after filtration through 0.22 µm filter)—SP/SE-F;(c)Salivary protein solution/skin extract after incubation (37 °C, 5 min)—SP/SK-I;(d)Salivary protein solution/skin extract filtrate (after filtration through 0.22 µm filter)—SP/SK-F;

Prior to electrophoretic analysis, the samples were dissolved in sample buffer [0.055 M Tris-HCl (pH = 6.8), 2% (*w*/*v*) SDS, 7% (*v*/*v*) glycerol, 0.0025% (*w*/*v*) bromophenol blue and 5% β-mercaptoethanol]. All samples were mixed with sample buffer in a 1:1 (*v*/*v*) ratio, except for the precipitates (c and f). The precipitates, obtained after centrifugation of the salivary protein solution/wine mixtures and salivary protein solution/skin or seed mixtures and removal of the supernatant, were reconstituted in 500 µL of sample buffer, stirred with a mechanical shaker for 1 h, and centrifuged before loading into the wells. For all samples, 75 µL was loaded into the wells. Upon completion of analysis, the gels were stained with Coomassie blue dye for 45 min, then destained, scanned, and analyzed using SigmaGel software (SigmaGel software version 1.1, Jandal Scientific, San Rafael, CA, USA).

### 2.5. Untargeted and Targeted UHPLC-QTOF-MS Analysis

An Agilent 1290 Infinity ultra-high-performance liquid chromatography (UHPLC) system coupled to a quadrupole time-of-flight mass spectrometry (6530C QTOF-MS) (Agilent Technologies, Inc., Santa Clara, CA, USA) was used to analyze the wine phenolic compounds. The applied HPLC method (mobile phase, gradient elution, flow rate, and injection volume) and the operating parameters of QTOF-MS were previously described in detail by Milinčić et al. [41]. The chromatographic separation was performed at a temperature of 40 °C on a Zorbax C18 column (2.1 × 50 mm, 1.8 µm) (Agilent Technologies, Inc., CA, USA). The QTOF-MS system was equipped with a Dual Agilent Jet Stream electrospray ionization (ESI) source, operating in both positive (ESI+) and negative (ESI−) ionization modes. Anthocyanins were analyzed in positive ionization mode, while flavan-3-ols, procyanidins, and other phenolic were monitored in negative ionization mode.

The young and aged Prokupac wines were analyzed in auto MS/MS acquisition mode (untargeted analysis) to gain more detailed insight into their phenolic profiles and evaluate differences/similarities between these wines. The parameters for the auto MS/MS mode were as follows: mass range (100–1700 *m*/*z*), acquisition rate (1 spectra/s), and acquisition time (1000 ms/spectrum). The phenolic compounds in young and aged wines were identified based on their monoisotopic mass and MS fragmentation patterns and confirmed by available standards or data from the literature [26,27,28,29,30]. The exact masses of the components were calculated using ChemDraw software (version 12.0, CambridgeSoft, Cambridge, MA, USA). Quantification was performed for selected wine phenolic compounds for which standards were available, and the content of each compound was expressed as mg/L wine. Phenolic standards were purchased from Chem Faces, with purity >98% (Wuhan, Hubei, China). The equation parameters, correlation coefficient (R2), limit of quantification (LOQ), and limit of detection (LOD) of the applied phenolic standards for quantification are shown in Appendix A.

After the saliva test, typical anthocyanins (malvidin derivatives), flavan-3-ols (catechin and epicatechin), and procyanidins (procyanidin dimer to pentamer) were selected and monitored via targeted UHPLC-QTOF-MS analysis to obtain more information about the chemical affinity of these compounds for salivary proteins. Targeted analysis is more sensitive than untargeted analysis and can be applied to detect predominant or trace compounds in the sample. The percentage of individual anthocyanins and procyanidins bound to salivary proteins was calculated as the ratio of the areas of target compounds in the filtrate (SP/YW-F and SP/AW-F) and control (CYW and CAW) samples. Data evaluation and analysis were performed using Agilent MassHunter software.

### 2.6. Sensory Analysis

This study involved sensory evaluation of wine samples conducted by twelve trained adult panelists (6 men and 6 women) from the Faculty of Agriculture, Belgrade, Serbia. The sensory evaluation included tastings of the investigated young and aged Prokupac wines. The panelists were selected on the basis of their interest, availability, and experience in sensory analysis. The evaluation did not involve any invasive procedures, collection of sensitive personal data, or commercial interests. All participants were fully informed about the nature of the study and voluntarily provided their written consent prior to participation, with the right to withdraw at any time. In accordance with the Code of Professional Ethics of the University of Belgrade, adopted by the Senate of the University of Belgrade and published in the Official Gazette of the Republic of Serbia, No. 189/16, p. 16, and considering the non-invasive nature of the sensory analysis, this study was exempt from ethical committee approval.

Sensory analysis took place in a thermo-regulated room at 20 °C and controlled humidity, according to ISO 8589:2007 standards [42], in individual booths. For each test, 15 mL of wine was presented in a colored glass, according to ISO 3591:1977 [43], coded with a three-digit number. The panelists rated the intensity of various descriptors on a 10-point scale (0–9), with 0 being the lowest and 9 being the highest intensity. The descriptors rated included the following mouthfeel attributes: acidity, bitterness, astringency, and tannin quality. In addition, the sensory evaluation of wines was also performed using Boxbaum’s model of positive rating with a maximum of 20 points on the following four sensory characteristics: color, clearness, aroma, and taste [44].

### 2.7. Statistical Analysis

All data were statistically analyzed using Microsoft Excel (Microsoft 365 version). For normally distributed data with homogeneous variances, Tukey’s post hoc parametric test was applied to determine the presence and degree of significant differences. One-way ANOVA was used for statistical analysis of the sensory analysis data. Statistical significance was set at a *p*-value < 0.05.

## 3. Results and Discussion

### 3.1. Untargeted UHPLC-QTOF-MS Profile of Young and Aged Prokupac Wines

The untargeted analysis of phenolic compounds in young and aged Prokupac wines yielded a total of 51 compounds, as shown in Table 1. All identified compounds were divided into several groups to gain a better insight into the differences/similarities between the analyzed wine samples: (I) phenolic acids (13 compounds); (II) flavan-3-ols and procyanidins (8 compounds); (III) flavonols and other flavonoids (15 compounds); (IV) stilbenoids (2 compounds); and (V) anthocyanins and derivatives (13 compounds).

Phenolic acids were detected mainly in the form of aglycones, ethyl derivatives, or esters with tartaric acid. Gallic and vanillic acids were the only hydroxybenzoic acids confirmed. Gallic acid was detected in both wines. However, vanillic acid was found only in young wine and its absence in APW was probably due to transformations (polymerization or oxidation) caused by the maturation of the wine. Coutaric acid, caftaric acid, and fertaric acid were confirmed in both wines and are typical hydroxycinnamoyltartaric acid derivatives identified in grapes and wine [4,45]. On the other hand, coumaric and ferulic acids were selectively detected in the analyzed wines, while caffeic acid was found in both samples. The presence of hydroxycinnamic acid in APW may be due to the hydrolysis of hydroxycinnamoyltartaric acid derivatives and caffeoylquinic acid (only present in YPW) that occurs during aging [35]. But a prolonged aging period may also affect the loss or absence of phenolic acids in aged wine [35], as in the case of coumaric acid. Ethyl gallate and ethyl caffeate were detected in both wine samples. Ethyl derivatives of phenolic acids are typical compounds in products undergoing alcoholic fermentation [4]. Ellagic acid was found in both wines. Previous studies have shown a high content of ellagic acid in Prokupac grape seeds, which apparently passes easily into the wine during fermentation. However, ellagic acid can also be extracted from oak barrels and is a characteristic marker for wine that has undergone barrique maturation [46,47], which explains the doubled content of these compounds in APW (Appendix A).

The main flavan-3-ols (catechin and epicatechin) and procyanidin dimer B-type isomers (with the exception of compound 17) were detected in both young and aged wines, although their levels were significantly higher in young wine (Appendix A). On the other hand, chalcan-flavan 3-ol dimer isomers and the procyanidin dimer B-type gallate were confirmed only in YPW. The absence of these compounds, as well as the lower content of procyanidin dimers in APW, can be caused by oxidative flavanol (interflavan) polymerization and aldehyde-mediated polycondensation of anthocyanins and flavanols during wine aging [35,48].

Flavonols in wine usually originate from the grape skins [45,49] and are present in the form of aglycones, hexuronide, or glycoside (hexoside). Of the flavonol aglycones, quercetin, myricetin, and isorhamnetin were confirmed in both wines, but their content was significantly reduced in APW (1.3–8.6 times). Other aglycones were found selectively in YPW (kaempferol and syringetin) or APW (laricitrin). Flavonol glycosides were most frequently found in YPW, while their proportion in APW was significantly lower or they were completely absent (laricitrin 3-*O*-hexoside). Flavonol hexuronides were also found in the wine samples, quercetin 3-*O*-hexuronide was found in both wines, and myricetin 3-*O*-hexuronide was found only in APW. These results were in agreement with the results reported by Gutierrez et al. [35], who analyzed young and shortly aged red wines. The reduction in flavonol glycosides may be caused by hydrolysis of the glycosidic linkage and formation of aglycones (as in the case of laricitrin) or by oxidation to hexuronides. In addition, glucosides and aglycones can also be involved in other oxidative reactions and condensation during wine aging [35]. Of the other non-anthocyanin flavonoids, naringenin was detected in both wines, while other phenolics were found selectively in YPW (dihidromyricetin and phlorizin) or APW (taxifolin).

Stilbenoids are widely present in wine [45,50] and grape stems [4], especially resveratrol, which is known for its health benefits [50]. Resveratrol and its hexoside (piceid) were confirmed in both wines. The amounts of resveratrol hexoside were similar in young and aged wines, but the resveratrol content was significantly higher in young wine (Appendix A).

Anthocyanin derivatives are responsible for the purple-red color of young wine, while the brick-red color of aged wine comes from various pyranoanthocyanins (anthocyanin-derived pigments) formed during fermentation and aging [34,51]. The presence or absence of anthocyanins and pyranoanthocyanins clearly showed differences between young and aged Prokupac wines (Table 1). Malvidin and pyranomalvidin derivatives were predominant in the wines analyzed and were most frequently confirmed. Malvidin 3-*O*-glucoside, vitisin A, and vitisin B were found in both wines, but these compounds were predominant found in young wine (Appendix A). In addition, malvidin-3-*O*-(6″-acetyl)-hexoside, malvidin-3-*O*-(6″-*p*-coumaroyl)-hexoside, and their pyrano derivatives (vitisin B-type compounds, 42 and 47, Table 1) were only detected in YPW. These vitisin-like pyranoanthocyanins were probably formed during fermentation by the reaction of free anthocyanins and certain yeast byproducts, such as acetaldehyde and pyruvic acid [34,45]. Previous studies have shown that the content of these vitisin-like pyranoanthocyanins decreases during wine aging in the bottle, which is due to the formation of various condensation products, as well as the absence of air/oak compounds that favor their formation and protect them from degradation [51]. Other anthocyanins detected, such as peonidin and petunidin glucosides and acetyl and/or coumaroyl derivatives were confirmed only in YPW (Table 1). Compounds recognized as malvidin-3-*O*-hexoside-4-vinylphenol (*m*/*z* 609, with major fragment at 447 *m*/*z*) and malvidin-3-*O*-hexoside-4-vinylcatechol (Pinotin A) (*m*/*z* 625, with major fragment at 463 *m*/*z*) were detected only in APW. These compounds were formed by different mechanisms during maturation in the barrel and during aging in the barrel and bottle and represent typical compounds in aged wine [51].

In order to ensure a better interpretation and understanding of the interactions between phenolic compounds and salivary proteins, the structural formulas of representative flavan-3-ols, procyanidins, and anthocyanins, confirmed by untargeted analysis of young and aged Prokupac wines, are presented in Figure 1.

### 3.2. Content of Phenolic Compounds in Young and Aged Prokupac Wines

The results of the quantification of phenolic compounds in young and aged Prokupac wines are shown in Table 2. The total content of all quantified phenolic compounds was 132.45 mg/L for YPW and 55.56 mg/L for APW. The contents of individual phenolic compounds (except ellagic acid and ferulic acid) were also significantly higher in YPW than in APW. The lower contents of monomeric phenolic compounds in APW were likely due to oxidation, copigmentation, polymerization, and condensation of these compounds during aging of the wine. Among the quantified phenolics, malvidin-3-*O*-glucoside was the most abundant in YPW (47.20 mg/L), followed by gallic acid (24.65 mg/L), catechin (16.32 mg/L), procyanidin dimers (18.91 mg/L), and epicatechin (6.93 mg/L). The contents of these compounds were significantly lower in APW, especially malvidin 3-*O*-glucoside (about 8-fold lower), which was attributed to its sensitivity and tendency to form pyranoanthocyanins and complexes with other phenolics during maturation and aging. In APW, gallic acid (19.84 mg/L) and catechin (11.40 mg/L) were the predominant compounds. Apart from the aforementioned compounds, other quantified phenolic compounds were either less abundant (<6 mg/L), present in trace amounts (<LOQ), or completely absent in both wines.

### 3.3. SDS-PAGE Analysis of Salivary Proteins Before and After Interaction with Wine Samples

To understand the binding affinity and mechanism of interactions between phenolic compounds and salivary proteins, and the tendency to form complexes between them, SDS-R-PAGE analysis of the salivary proteins was performed before and after mixing with wine/skin/seed samples (Figure 2a,b).

The salivary proteins were identified based on the literature data [17,18,21], with several predominant and/or diffuse bands corresponding to mucins (<95 kDa), amylase (~62 kDa), proline-rich proteins (PRPs), cystatins (10–14 kDa), and statherins (6.5–10 kDa) (Figure 2a,b, lines CSP). The polypeptide composition (%) of the salivary proteins is shown in Appendix A. Proline-rich proteins (acidic and basic PRPs), representing the main fraction of salivary proteins, had a share of 50.52% (Appendix A). Bands of proline-rich proteins can be observed in two regions with the following MW ranges: (a) 14 to 37 kDa (acidic and basic PRPs) and (b) 66–95 kDa (weakly glycosylated PRPs). Previous studies have shown that proline-rich proteins and statherins are the most prone to reacting with phenolics, especially with acidic PRPs [18,29]. The electrophoretic patterns of the salivary protein/wine filtrates (SP/YW-F and SP/AW-F) were empty, with no visible bands originating from salivary proteins (Figure 2a). This meant that salivary proteins were retained in the precipitates and were present as insoluble complexes with phenolic compounds and rarely in free form, as shown in the SP/YW-P and SP/AW-P patterns (Figure 2a). Five intensive bands, with molecular masses of 75.9, 61.9, 58.3, 13.1, and 11.7 kDa (Figure 2a, red marked numbers, 3, 4, 5, 19, and 20), can be observed in the electrophoretic patterns of both precipitates (Figure 2a, lines SP/YW-P and SP/AW-P). These bands (especially bands 4, 19, and 20) showed similar electrophoretic pathways as some salivary proteins (see line CSP, Figure 2a), but these bands were more intensive compared to the salivary bands and could be attributed to the formation of phenolics/salivary protein complexes. Bands 3, 4, and 5 (Figure 2a) could be attributed to complexes between phenolics and acidic or basic PRPs, while bands 19 and 20 (Figure 2a) were probably complexes of phenolics and histatins or statherins (Figure 2a, see lines SP/YW-P and SP/AW-P) [17,18,21,29,30]. These bands (except band 3) were more intense for AW/SP-P than the same bands in the SP/YW-P pattern (Figure 2a), especially band 5. These differences in band intensity were probably due to the different abilities of phenolics in young and aged wines to react with salivary proteins. Procyanidins, interflavan or flavanol/anthocyanin polymers, and ellagitannins in aged wine obviously showed a high affinity to bind to salivary proteins and form insoluble complexes [8,19,22,38]. On the other hand, procyanidins with a high affinity for salivary proteins [30], as well as monomeric flavan-3-ols (catechin and epicatechin) and anthocyanins which preferentially form soluble aggregates [26,33,52], were predominantly detected in young wine, while polymeric phenolics were less represented or completely absent. Furthermore, in both precipitates (Figure 2a, lines SP/YW-P and SP/AW-P), numerous diffuse bands of low intensity in the MW range from 16 to 52 kDa could be observed. These bands could also be associated with complexes formed between wine phenolics and salivary proteins. A band of high-molecular-weight complexes could be noticed at the entrance to the upper gel for both precipitates (Figure 2a). These complexes were probably formed by phenolics and glycosylated PRPs [20]. As expected, no bands were visible in the electrophoretic patterns of CYPW and CAPW (Figure 2a).

To ensure a better visualization of the newly formed complexes and the decreasing/increasing band intensity of individual salivary proteins after interaction with wine phenolics, electrophoregrams of CSP and precipitates (SP/YW-P and SP/AW-P) are presented in Figure 3a. As can be seen, the peaks of acidic and basic PRPs on SP/YW-P and SP/AW-P electrophoregrams were significantly reduced or absent compared to the same peaks on the CSP electrophoregram, indicating that these fractions of salivary proteins were crucial for the formation of complexes with phenolic compounds. On the other hand, several high-intensity (3, 4, 5, 19, and 20, Figure 3a) and low-intensity (6, 11, 13, 16, 18, and 22, Figure 3a) peaks could be seen in the SP/YW-P and SP/AW-P electrophoregrams, probably originating from newly formed complexes, as previously observed (Figure 2a).

The previous observations are also summarized in Table 3, which shows the changes (%) in the contents of the individual salivary proteins in the CSP and in the precipitates (SP/YW-P and SP/AW-P), as all salivary proteins and complexes were retained in the precipitates after interaction. As can be seen, the previous observations were in agreement with the results of the densitometric analysis (Table 3).

To determine the interactions between phenolics and salivary proteins, the type of aggregates/complexes formed (soluble or insoluble), and to confirm the observations made previously, a control saliva test was also carried out. This experiment was concerned with the interaction of salivary proteins with grape seed and grape skin extracts, which predominantly contained flavan-3-ols/procyanidins or anthocyanins, respectively. The electrophoretic patterns of SP/SK-I and SP/SK-F were similar, with the same electrophoretic pathways and mobility of all identified salivary peptides and/or complexes observed, which were not disrupted under reducing conditions (Figure 2b). This indicated the low ability of anthocyanins and other monomeric grape skin phenolics (flavonols) to interact with salivary proteins and form soluble aggregates [24,26,33], as can be observed in the SP/SE-F pattern. On the other hand, in the SP/SE-I electrophoretic patterns, interactions could be observed mainly between grape seed procyanidins and salivary proteins, especially acidic/basic PRPs (Figure 2b). However, only several bands of low intensity (around 66 kDa) were visible in the electrophoretic SP/SE-F pattern, demonstrating the tendency of grape seed procyanidins to form insoluble complexes with salivary proteins [8,17,18,19,21]. The electrophoretic patterns of the filtrates (SP/YW-F and SP/AW-F) (Figure 2a) and SP/SE-F (Figure 2b) were similar, as were the electrophoretic patterns of precipitates (SP/YW-P and SP/AW-P) (Figure 2a) and SP/SE-I (Figure 2b). This indicated that the procyanidins and the polymeric forms of flavan-3-ols present in wine were key to the interaction with salivary proteins and the formation of oral sensations. In addition, the differences between the complexes formed (insoluble complexes or soluble aggregates) between the salivary proteins and the procyanidins from the seeds or anthocyanins from the skins could be clearly seen in the electrophoregrams shown in Figure 3b.

### 3.4. Binding Affinity of Salivary Proteins for Selected Wine Phenolics

Targeted analysis of selected procyanidins and anthocyanins before and after interaction with salivary proteins revealed their individual affinities to bind to salivary proteins and their contributions to sensory perceptions and astringency. As can be seen, (epi)catechin, procyanidins, and anthocyanins in young and aged Prokupac wines showed different chemical affinities to salivary proteins (Table 4).

Flavan-3-ols and all procyanidin oligomers (from dimer to pentamer) showed a tendency to bind to salivary proteins. The lowest binding affinity was observed for (epi)catechin, while the binding ability of procyanidins increased from dimer to pentamer in both wines. These results were in agreement with the observations of other studies [8,19,30,31,32] investigating the interactions between grape seed/wine procyanidins and salivary proteins. Similar to our results, Ma et al. [8] showed that larger procyanidins oligomers (trimers, tetramers, and pentamers) had a stronger affinity for salivary proteins. The increased binding affinity of the procyanidin pentamer and other oligomers may be attributed to their enhanced ability to form multiple hydrogen bonds and hydrophobic interactions (Figure 4a) with salivary proteins, mainly PRPs [53]. However, the percentages of bound (epi)catechin and procyanidins were significantly higher in aged Prokupac wine (54.00% epicatechin to 100% procyanidin pentamer) than in young wine (4.78% epicatechin to 32.16% procyanidin pentamer). The transformation of procyanidins during wine aging probably affects their composition and structure, which could increase their binding efficiency for salivary proteins. However, the binding affinities of tannins (polymerized forms) in aged wine for polyprolines were variable and depended on the vintage and number of years of aging of the wine [53].

Interestingly, the results of the binding affinity of anthocyanins in young and aged wines showed significant differences. Anthocyanins in young wine showed little or no binding affinity for salivary proteins (0 to 15.86%). Previous studies have also indicated that poor electrostatic (ionic) interactions between anthocyanins and salivary proteins [24,33] and the formation of soluble aggregates (Figure 4a) contribute to the perception of astringency [26]. On the other hand, the same anthocyanins and vitisin B in aged wine were almost completely retained in the precipitate, apparently showing a “high” binding affinity. This can also be explained by the fact that the contents of total and individual anthocyanins in APW were significantly lower than in YPW (Appendix A). For example, the content of malvidin 3-*O*-glucoside was almost 8-fold lower in APW than in YPW (Table 2). As aforementioned, these differences in the binding affinities of selected phenolics (epicatechin, procyanidins, and anthocyanins) can be explained by the different compositions of young and aged wines (Table 1 and Table 2), as well as by the synergistic effects of other wine phenolics [33]. In the case of aged Prokupac wine, procyanidins, high-molecular-weight tannins, and ellagitannins showed high binding affinities for salivary proteins [19,21,30,38] and formed insoluble complexes that probably collected and intensively bound other phenolics during precipitation. The results of the targeted analysis supported the conclusions and results of the electrophoretic analyses.

Taking into account previous interpretations [33,54,55,56,57] and the results of this study, the mechanism of interactions between proline-rich proteins and the major wine phenolics is illustrated (Figure 4a), along with a schematic representation of the formation of insoluble complexes between PRPs and phenolic compounds in young (Figure 4b) and aged (Figure 4c) wines.

### 3.5. Sensory Analysis

The results of the sensory analysis are summarized in Figure 5, which visually highlights the perceptual differences between young and aged Prokupac wines. Aged wine was characterized by higher scores in tannin quality, while young wine showed greater acidity and astringency. Bitterness appeared comparable between the two samples. These sensory trends were further supported by statistical analysis (Appendix A). Statistically significant differences between young and aged Prokupac wines were indicated for the parameters acidity, astringency, and tannin quality, while the *p*-value for bitterness showed no significant differences. The sensorial analysis of the young and aged Prokupac wines revealed differences between the wines, but not between the panelists involved in the study.

The increased perception of tannin quality in aged wine could be attributed to the polymerization and structural changes of tannins over time, which alter their interactions with salivary proteins. These findings were consistent with the results of the electrophoretic and targeted UHPLC-QTOF-MS analyses, which showed more intensive interactions between aged wine phenolics and salivary proteins, especially procyanidins and other polymeric molecules. In addition, soluble complexes of anthocyanin glucosides and acyl derivatives were recognized as carriers of bitterness [24,28], but the sensory analysis showed that there were no differences in bitterness between young and aged wines. By contrast, the proportion of bound anthocyanins in aged wine was obviously high and was caused by their precipitation with insoluble tannin-procyanidin-ellagitannin/salivary protein complexes. Finally, the lack of variation between panelists indicated a consistent sensory perception, which increased the reliability of the results. These results contribute to a deeper understanding of the development of Prokupac wine and its impact on consumer perception.

To complement the statistical findings, a descriptive sensory evaluation based on Boxbaum’s model was conducted, further illustrating the sensory distinctions between the wines (Appendix A). Young wine exhibited a closed red color and slight opalescence due to the presence of colloids, with moderate fluidity in the glass. The aroma showed signs of slight degradation of aromatic compounds along with notes of overripe berry notes (bereton). The overall impression was clean, varietal, and typical, with moderate intensity. On the palate, the wine was moderately full-bodied, lacking balance in the finish due to the cumulative effect of acidity and tannins with moderate persistence. Aged wine displayed a closed ruby color with clarity. The aroma was clean, of moderate intensity, evoking ripe red fruit. On the palate, the wine showed a well-balanced interplay of alcoholic sweetness, tannins, and acidity, with a moderately persistence aroma.

## 4. Conclusions

This work provides valuable insight into the interaction of red wine phenolics with salivary proteins and highlights the role of procyanidins and anthocyanins in modulating mouthfeel sensations such as astringency and bitterness. This is the first time that young and aged Prokupac wines have been studied in this area of scientific research, representing a new discovery. As expected, UHPLC-QTOF-MS profiling of the wines showed fundamental differences in phenolic complexes between young and aged wines. Crucial findings obtained by SDS-PAGE confirmed that procyanidins are the primary drivers of salivary protein binding and form insoluble complexes with salivary proteins. By contrast, anthocyanins form complexes mainly in aged Prokupac wine, suggesting that anthocyanins co-precipitate with insoluble complexes of polymerized procyanidins and salivary proteins. The overall results of the targeted UHPLC-QTOF-MS analysis (ESI+/ESI-) confirmed that wine aging affects these interactions, with aged Prokupac wine having a higher affinity for salivary proteins due to an increased presence of polymerized procyanidins, complex pigments, and ellagitannins. Complementary sensory evaluation confirmed these biochemical findings, showing that aged wine was perceived as having higher tannin quality, while young wine was characterized by greater acidity and astringency, with the bitterness being comparable. The results of this research expand our understanding of the molecular affinity of wine phenolics to salivary proteins and open up new possibilities for research into the sensory properties of red wine.

## Figures and Tables

**Figure 1 foods-14-01780-f001:**
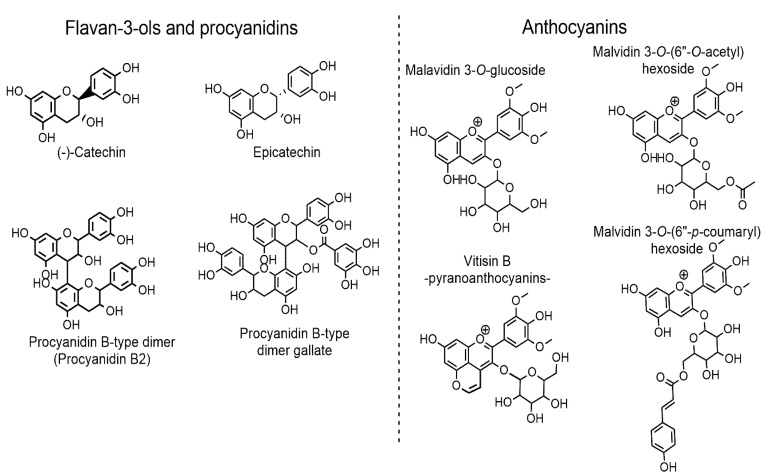
Structural formulas of main flavan-3-ols, procyanidins, and anthocyanins/pyroanthocyanin confirmed in young and/or aged Prokupac wines by untargeted analysis.

**Figure 2 foods-14-01780-f002:**
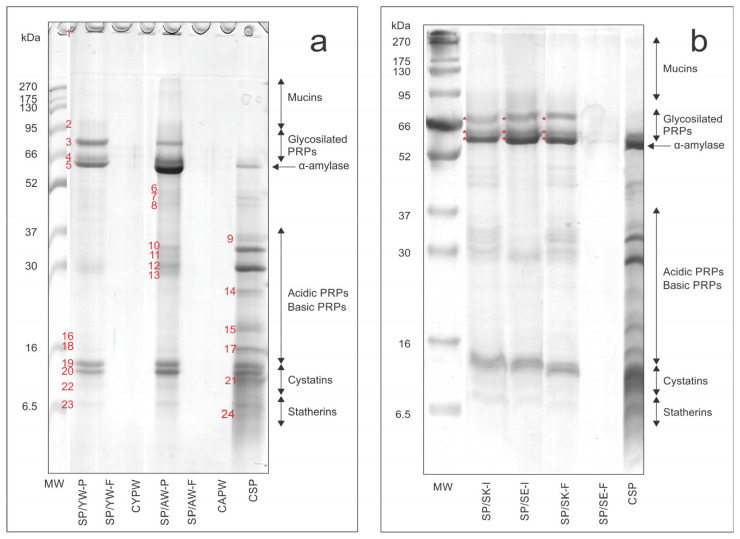
Electrophoretic patterns of salivary proteins before and after interaction with: (**a**) young and aged Prokupac wines and (**b**) Prokupac grape seed and skin extracts, analyzed by SDS-PAGE under reducing conditions (SDS-R-PAGE). Abbreviations are explained in Section 2.4; MW—molecular weight standard; PRPs—proline-rich proteins. Red numbers on the electrophoretic patterns of (**a**) mark bands of salivary proteins and complexes formed after interaction with phenolics. Red * on the electrophoretic patterns of (**b**) mark bands of complexes, GPRPs and/or complexes, and α-amylase and/or complexes, respectively.

**Figure 3 foods-14-01780-f003:**
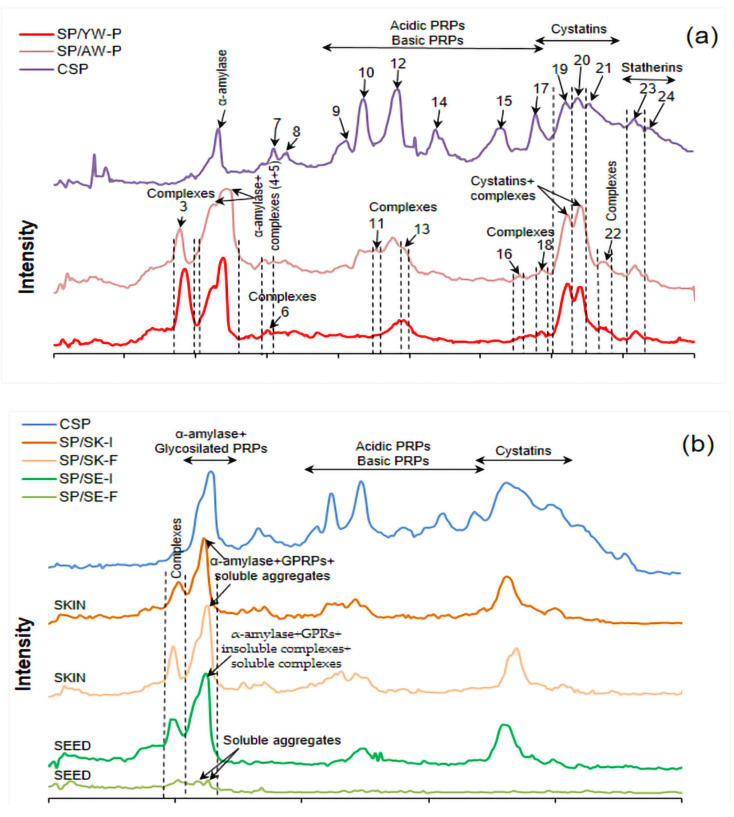
Electrophoregrams of: (**a**) salivary proteins (CSP) and precipitates after interaction with young and aged wines (SP/YW-P and SP/AW-P); (**b**) salivary proteins (CSP), CSP/seed and CSP/skin fractions after incubation (SP/SE-I and SP/SK-I), and CSP/seed and CSP/skin filtrates (SP/SE-F and SP/SK-F). Abbreviations are also explained in Section 2.4; PRPs—proline-rich proteins. The numbers mark peaks (Figure 3a) in accordance with the numbers marked in Figure 2a.

**Figure 4 foods-14-01780-f004:**
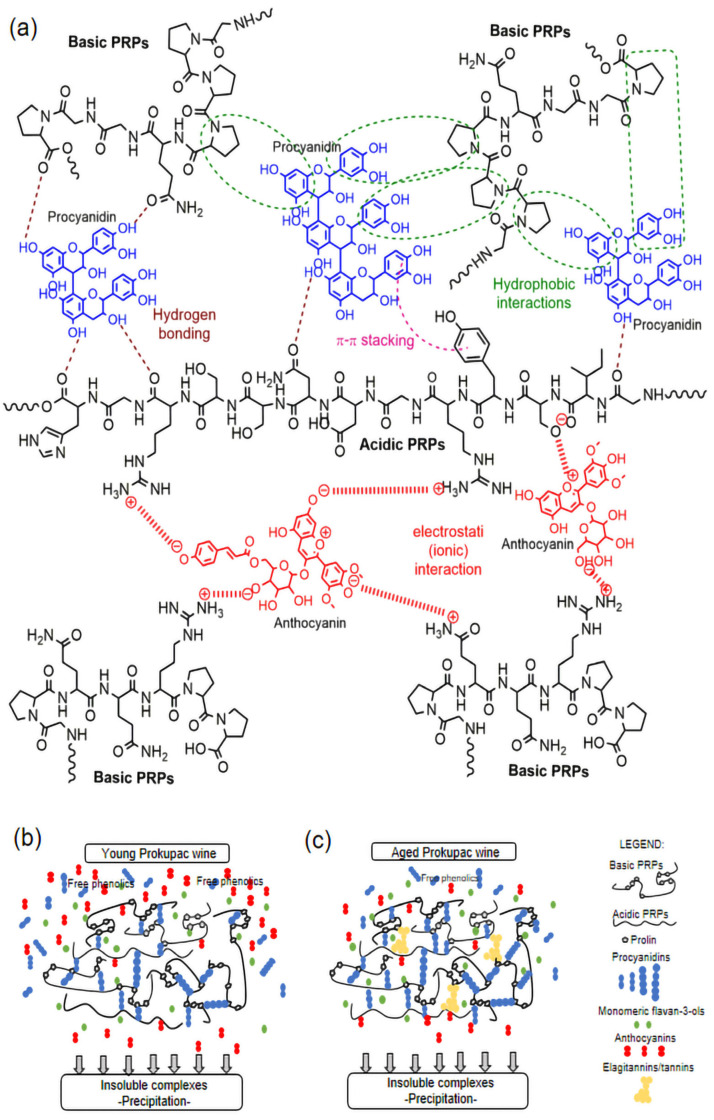
A schematic representation of: (**a**) interactions between proline-rich proteins and procyanidins/anthocyanins; (**b**) formation of insoluble complexes between PRPs and phenolic compounds in young wine; and (**c**) formation of insoluble complexes between PRPs and phenolic compounds in aged wine.

**Figure 5 foods-14-01780-f005:**
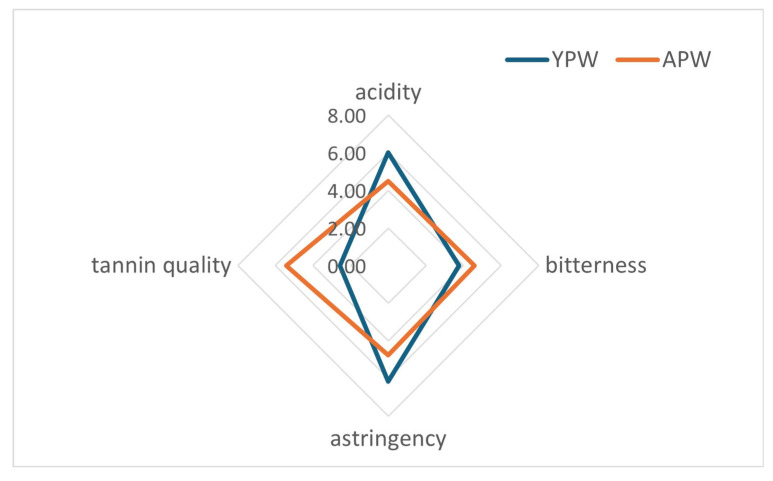
Sensory analysis of young Prokupac wine (YPW) and aged Prokupac wine (APW).

**Table 1 foods-14-01780-t001:** Untargeted UHPLC-QTOF-MS phenolic profile of young and aged Prokupac red wines.

No.	RT	Compound	Formula	Calculated Mass	*m/z* Exact Mass	mDa	MS Fragments (Main Fragment)	Samples	
								YPW	APW
Phenolic acid and derivatives									
1	**2.80**	Coumaric acid *	C_9_H_7_O_3_^−^	163.0395	163.0401	0.58	**119.0497(100)**	+	−
2	7.38	Vanillic acid *	C_8_H_7_O_4_^−^	167.0344	167.0356	1.17	**123.0439(100)**, 107.0133	+	−
3	1.00	Gallic acid *	C_7_H_5_O_5_^−^	169.0137	169.0148	1.10	**125.0239(100)**, 124.0163	+	+
4	4.37	Caffeic acid *	C_9_H_7_O_4_^−^	179.0344	179.0356	1.17	**135.0445(100)**, 134.0371, 107.0499	+	+
5	3.92	Ferulic acid *	C_10_H_9_O_4_^−^	193.0501	193.0503	0.22	**134.0365(100)**, 133.0283, 117.0342, 148.0133, 164.0119	−	+
6	6.59	Ethyl gallate	C_9_H_9_O_5_^−^	197.045	197.0465	1.50	**124.0162(100)**, 125.0227, **169.0144**	+	+
7	9.42	Ethyl caffeic acid	C_11_H_11_O_4_^−^	207.0657	207.0670	1.27	**133.0292(100)**, 135.0446, 134.036, 161.0244, **179.0343**	+	+
8	3.20	Coutaric acid	C_13_H_11_O_8_^−^	295.0454	295.0470	1.61	**119.0501(100)**, **163.0400**	+	+
9	7.52	Ellagic acid *	C_14_H_5_O_8_^−^	300.9984	301.0001	1.66	**300.9992(100)**, 299.9913, 283.9966, 229.016, 201.0202, 151.0033, 245.0144, 185.0251, 173.0229, 257.0103	+	+
10	1.54	Caftaric acid	C_13_H_11_O_9_^−^	311.0403	311.0421	1.79	**135.0447(100)**, 149.0089, **179.0352**, 134.0372	+	+
11	4.17	Fertaric acid	C_14_H_13_O_9_^−^	325.056	325.0600	4.04	**134.0368(100)**, **193.0506**, 178.027, 149.0089	+	+
12	7.81	Aesculin	C_15_H_15_O_9_^−^	339.0716	339.0734	1.79	**161.0241(100)**, 159.0295, 133.0285, **177.0398**, 115.0392	−	+
13	3.84	Caffeoylquinic acid (like Chlorogenic acid)	C_16_H_17_O_9_^−^	353.0873	353.0887	1.44	**191.0559(100)**, 161.0239, 127.0395, 173.0451, **135.0449**	+	−
Flavan-3-ols and procyanidins									
14	3.42	Catechin *	C_15_H_13_O_6_^−^	289.0712	289.0727	1.49	**123.045(100)**, 109.0294, 125.0244, 151.0398, 137.0244, 203.0712, 149.025, 221.0821, 187.0402, 245.0813	+	+
15	6.13	Epicatechin *	C_15_H_13_O_6_^−^	289.0712	289.0727	1.49	**123.045(100)**, 109.0294, 125.0244, 151.0399, 137.0243, 203.0713, 149.0253, 221.0819, 187.0403, 245.0820	+	+
16	2.48	Procyanidin B-type dimer is. I	C_30_H_25_O_12_^−^	577.1346	577.1365	1.90	**289.0724(100)**, **407.0780**, 125.0243, 245.0805, 161.0248, 137.0242, 273.0408, **425.0884**, **451.1036**, 255.0339, 229.0511	+	+
17	4.11	Procyanidin B-type dimer is. II	C_30_H_25_O_12_^−^	577.1346	577.1365	1.90	**289.0718(100)**, **407.0776**, 125.0241, 245.0798, 161.0249, 137.0239, 273.0404, **425.0885**, **451.1047**, 255.0377, 229.0512, 205.0485	+	−
18	5.38	Procyanidin B-type dimer is. III	C_30_H_25_O_12_^−^	577.1346	577.1365	1.90	**289.0722(100)**, **407.0778**, 125.0242, 245.0803, 161.0250, 137.0242, 273.0407, **425.0882**, **451.1031**, 229.0512, 205.0476, 109.0291	+	+
19	3.41	Chalcan-flavan 3-ol dimer is. I (like Gambiriin A1)	C_30_H_27_O_12_^−^	579.1503	579.1522	1.95	**289.0720(100)**, 245.0824, 271.0607, 179.0352, 205.0510, 165.0187, 151.0400, 137.0245, 125.0242, 109.0293	+	−
20	6.07	Chalcan-flavan 3-ol dimer is. II	C_30_H_27_O_12_^−^	579.1503	579.1522	1.95	**289.0719(100)**, 245.0824, 271.060719, 179.0352, 205.0510, 165.0188, 151.0397, 137.0241, 125.0241, 109.0293, 221.0825	+	−
21	6.84	Procyanidin dimer B-type gallate	C_37_H_29_O_16_^−^	729.1456	729.1481	2.54	**407.0772(100)**, **289.0716**, 125.0239, 451.1023, **169.0141**, 577.1319, 271.0612, 287.0567, 441.0825, 161.0246, 245.0591, 203.0206	+	−
Flavonols and glycosides									
22	10.1	Kaempferol *	C_15_H_9_O_6_^−^	285.0399	285.0411	1.19	**285.0405(100)**, 185.0609, 229.0515, 239.035, 159.0447, 211.0396, 143.0497, 151.0038, 227.0347, 255.0301, 268.0370	+	−
23	9.30	Quercetin	C_15_H_9_O_7_^−^	301.0348	301.0368	1.97	**151.0036(100)**, 121.0292, **178.9984**, 149.0237, 301.0334, 245.0456, 273.0400, 229.0500, 201.0549	+	+
24	10.3	Isorhamnetin	C_16_H_11_O_7_^−^	315.0505	315.0516	1.12	**300.0276(100)**, 151.0033, 301.031, 107.0133, 271.0251, 283.0259, 255.0293, 227.0344, 243.0301, 179.0001	+	+
25	8.41	Myricetin *	C_15_H_9_O_8_^−^	317.0297	317.0315	1.76	**151.0036(100)**, 137.0241, 107.0137, 178.9987, 165.0191, 227.0349, 243.0311, 271.0247, 317.0306	+	+
26	9.27	Laricitrin	C_16_H_11_O_8_^−^	331.0454	331.0473	1.91	**151.0062(100)**, 316.0231, 178.9995, 271.0243, **317.0257**, 287.0179, 259.0252, 243.0300, 107.0135	-	+
27	7.72	Syringetin	C_17_H_13_O_8_^−^	345.061	345.0634	2.36	**190.9994(100)**, **315.0144**, 163.0028, 287.0211, **330.0383**, 316.019, 271.0243, 259.0244, 243.0282, 345.0607	+	−
28	7.60	Quercetin 3-*O*-hexuronide	C_21_H_17_O_13_^−^	477.0669	477.0687	1.78	**301.0358(100)**, 151.0034, 178.9984, 283.0251, 273.0403, 255.0301, 245.0451	+	+
29	7.13	Myricetin 3-*O*-hexoside	C_21_H_19_O_13_^−^	479.0826	479.0847	2.13	**316.0229(100)**, 271.0245, 287.0194, 178.9982, 151.0035, 479.0832	+	+
30	7.05	Myricetin 3-*O*-hexuronide	C_21_H_17_O_14_^−^	493.0618	493.0647	2.87	**317.0304(100)**, 318.0312, 178.9971, 151.0049, 137.0232, 271.0281, 299.0174	−	+
31	7.65	Laricitrin 3-*O*-hexoside	C_22_H_21_O_13_^−^	493.0982	493.0988	0.58	**330.0382(100)**, 331.0446, **315.0150**, 316.0201, 287.02, 493.1013, 271.0245, 243.0285, 151.0055, 178.9975	+	-
32	8.11	Syringetin 3-*O*-hexoside	C_23_H_23_O_13_^−^	507.1139	507.1156	1.73	**344.0541(100)**, 345.0591, 507.1147, 273.0405, 301.0369, **316.0588**, **329.0321**, 258.0160, 151.0034	+	+
Other detected non-anthocyanin flavonoids									
33	9.83	Naringenin *	C_15_H_11_O_5_^−^	271.0606	271.0622	1.55	**119.0501(100)**, 151.0034, 107.0133, 177.0182, 161.0586, 145.0275, 229.0541	+	+
34	7.39	Taxifolin	C_15_H_11_O_7_^−^	303.0505	303.0522	1.72	**125.0249(100)**, 151.0216, 174.0312, 199.0390, 137.0211, 193.0515, 243.0271	−	+
35	5.05	Dihydromyricetin	C_15_H_11_O_8_^−^	319.0454	319.0469	1.51	**125.0242(100)**, 165.019, 151.0038, 167.0346, 137.0241, 175.0040, 193.0137, 205.0501, 233.0457	+	−
36	8.40	Phloretin 2’-*O*-hexoside (like Phlorizin)	C_21_H_23_O_10_^−^	435.1291	435.1316	2.48	**167.0351(100)**, **273.0778**, 125.0238, 274.0802, 179.0348, 123.0452, 168.0388	+	−
Stilbenoids									
37	9.34	Resveratrol *	C_14_H_11_O_3_^−^	227.0708	227.0721	1.28	**143.0501(100)**, 185.0593, 117.0347, 157.0655, 167.0535	+	+
38	8.22	Resveratrol hexoside (like Piceid)	C_20_H_21_O_8_^−^	389.1236	389.1253	1.66	**227.0711(100)**, 185.0605, 143.0499, 159.0811	+	+
Anthocyanins and pyranoanthocyanins									
Malvidin derivatives									
39	6.59	Malvidin 3-*O*-glucoside *	C_23_H_25_O_12_^+^	493.1346	493.1375	2.9	**331.0831(100)**, 332.0854, 315.0508, 316.0578, 287.0555	+	+
40	7.13	Malvidin 3-*O*-hexoside-acetaldehyde (Vitisin B)	C_25_H_25_O_12_^+^	517.1346	517.1367	2.1	**355.0819(100)**, 356.0854, 317.0662	+	+
41	7.40	Malvidin 3-*O*-(6”-acetyl)hexoside	C_25_H_27_O_13_^+^	535.1452	535.1475	2.33	**331.0819(100)**, 332.085, 333.0878, 315.0505	+	-
42	7.45	10H-Pyranomalvidin 3-*O*-(6”-acetyl)hexoside (Malvidin-acetaldehyde adduct I)	C_27_H_27_O_13_^+^	559.1452	559.147	1.83	**355.0822(100)**, 356.0848, 397.0921	+	−
43	7.12	Malvidin 3-*O*-hexoside-pyruvate (Vitisin A)	C_26_H_25_O_14_^+^	561.1244	561.1266	2.17	**399.0722(100)**, 400.0754	+	+
44	8.64	Malvidin 3-*O*-hexoside-4-vinylphenol	C_31_H_29_O_13_^+^	609.1608	609.1626	1.78	**447.1079(100)**, 448.1112, 431.0755	-	+
45	8.39	Malvidin 3-*O*-hexoside-4-vinylcatechol (Pinotin A)	C_31_H_29_O_14_^+^	625.1557	625.1577	1.97	**463.1026(100)**, 464.1059, 447.0745	−	+
46	8.22	Malvidin 3-*O*-(6”-*p*-coumaroyl)hexoside	C_32_H_31_O_14_^+^	639.1714	639.1739	2.52	**331.0819(100)**, 332.085, 333.0876	+	−
47	8.11	10H-Pyranomalvidin 3-*O*-(6”-*p*-coumaroyl)hexosid (Malvidin-acetaldehyde adduct II)	C_34_H_31_O_14_^+^	663.1714	663.1737	2.32	**355.0811(100)**, 356.0852, 357.087	+	−
Other detected anthocyanins									
48	6.06	Petunidin 3-*O*-glucoside	C_22_H_23_O_12_^+^	479.119	479.1205	1.55	**317.0657(100)**, 318.0698, 302.0423	+	−
49	7.59	Peonidin 3-*O*-(6”- acetyl)hexoside	C_24_H_25_O_12_^+^	505.1346	505.1362	1.6	**301.0704(100)**, 302.0746, 286.048	+	−
50	8.30	Peonidin 3-*O*-(6”-*p*-coumaroyl)hexoside	C_31_H_29_O_13_^+^	609.1608	609.1635	2.68	**301.0708(100)**, 302.0744, 303.076, 286.0477	+	−
51	8.06	Petunidin 3-*O*-(6”-*p*-coumaroyl)hexoside	C_31_H_29_O_14_^+^	625.1557	625.1581	2.37	**317.0661(100)**, 318.0689, 302.0466	+	−

Abbreviations: is.—isomer; “−” nonidentified compound; “+” identified compound; YPW—Young Prokupac wine; APW—Aged Prokupac wine. * Phenolic compound confirmed by available standards.

**Table 2 foods-14-01780-t002:** Contents of phenolic compounds (mg/L) in young and aged Prokupac red wines.

Compound	Samples	
	YPW	APW
	mg/L Wine	
Coumaric acid	3.07	–
Vanillic acid	5.95	–
Gallic acid	24.65	19.84
Caffeic acid	4.72	3.39
Ferulic acid	–	2.09
Ellagic acid	0.61	1.25
Resveratrol	1.38	<LOQ
Catechin	16.32	11.40
Epicatechin	6.93	2.74
Procyanidin B-type dimer is. I *	10.33	3.63
Procyanidin B-type dimer is. II *	2.02	–
Procyanidin B-type dimer is. III *	6.56	3.33
Kaempferol	<LOQ	–
Myricetin	2.69	2.09
Naringenin	<LOQ	<LOQ
Malvidin-3-*O*-glucoside	47.20	5.80
∑	132.45	55.56

* Expressed as procyanidin B2 equivalents. “–” nonidentified compounds; <LOQ—less then limit of quantification.

**Table 3 foods-14-01780-t003:** The changes (%) in individual salivary protein contents in the control saliva solution (CSP) and after interaction with wine phenolics (the same band was confirmed in the SP/AW-P and SP/YW-P electrophoretic patterns) retained in the precipitates.

No. Polypeptide Band	CSP (%)	SP/AW-P (%)	SP/YW-P (%)	Characteristic of Identified Bands	Band Area Ratio (SP/AW-P)/ (SP/YW-P)
2	−	+	+	Complexes	1.03
3	−	+	+	Complexes	0.65
4 + 5	100	643.4	346.9	α-amylase + GPRPs + complexes	1.85
6	−	+	+	Complexes	2.09
7	100	21.2	−	-	*
8	100	60.3	−	-	*
9	100	−	−	-	-
10	100	52.3	−	PRPs	*
11	−	+	+	Complexes	4.37
12	100	54.2	43.0	PRPs	1.26
13	−	+	+	Complexes	0.78
14	100	−	−	PRPs	-
15	100	−	−	PRPs	-
16	−	+	+	Complexes	0.93
17	100	−	−	PRPs	-
18	−	+	+	Complexes	1.60
19	100	150.4	137.0	Cystatins + complexes	1.09
20	100	120.7	75.8	Cystatins + complexes	1.59
21	100	−	−	Cystatins	-
22	−	+	+	Complexes	2.86
23	100	43.6	19.2	Statherins	2.27
24	100	0	0	Statherins	-

Abbreviations: Contents of the individual salivary proteins in the CSP given as 100%. “−” unidentified polypeptides; “+” identified polypeptide band only on SP/AW-P and SP/YW-P patterns (complexes). Band area ratio (SP/AW-P)/(SP/YW-P)—ratio of areas of the same polypeptide bands confirmed on SP/AW-P and SP/YW-P patterns. * Polypeptide confirmed in SP/AW-P electrophoretic pattern only.

**Table 4 foods-14-01780-t004:** Binding affinities (%) of salivary proteins for selected anthocyanins, flavan-3-ols, and procyanidins in young and aged Prokupac wines, analyzed by targeted UHPLC-QTOF-MS.

Target Compound	*m/z* Exact Mass	SP/YW	SP/AW
		Percentage of Bound Phenolics (%)	
Monomeric flavan-3-ol and procyanidins (ESI−)			
(Epi)catechin	289.0712	4.78 ± 0.95	54.00 ± 1.10
Procyanidin dimer (procyanidin B1)	577.1346	28.51 ± 0.85	71.25 ± 0.31
Procyanidin trimer (procyanidin C1)	865.1979	29.86 ± 1.94	77.87 ± 0.24
Procyanidin tetramer	1153.2614	23.44 ± 1.21	74.99 ± 0.95
Procyanidin pentamer	1441.3248	32.16 ± 3.37	100
Anthocyanins (malvidin derivatives) and pyranoanthocyanins (ESI+)			
Malavidin 3-O-glucoside	493.1346	3.91 ± 0.35	90.01 ± 0.03
Malvidin 3-O-(6″-O-acetyl)hexoside	535.1452	/	100
Malvidin 3-O-(6″-p-coumaroyl)hexoside	639.1714	15.86 ± 0.97	100
Malvidin 3-O-hexoside-acetaldehyde (Vitisin B)	517.1346	/	78.31 ± 0.48

Values are presented as the mean ± standard deviation, n = 3.

## Data Availability

The original contributions presented in this study are included in the article/Appendix A, and further inquiries can be directed to the corresponding authors.

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
