# Peer review of "Procyanidins and Anthocyanins in Young and Aged Prokupac Wines: Evaluation of Their Reactivity Toward Salivary Proteins"

_foods, 2025, doi:10.3390/foods14101780_

Round 1

Reviewer 1 Report

Comments and Suggestions for Authors

The paper entitled ‘Procyanidins and anthocyanins of young and aged Prokupac wines: Evaluation of their reactivity towards salivary proteins’ discussed influence of wine aging on the interaction between phenolic compounds and salivary proteins and emphasize the dominant role of procyanidins in protein binding and the potential synergistic contribution of anthocyanins to mouthfeel perception.

The manuscript is quite good as for contents, conclusions are clear and the Ref list is complete and correct.

While I don’t have any particular scientific issues with this paper, there are certain aspects of the manuscript that must be revised. Below are the specific details:

  1. Figure 1 displays the electrophoretic patterns of salivary proteins prior to and following interaction. It is crucial to note that a well-written manuscript should always be clear and comprehensible. At least, that is what I believe as a reviewer. Therefore, I highly recommend that the authors include a table to visually summarize this figure.
  2. To improve the readability of the paper, it would be beneficial to add the structural formulas of various anthocyanins, which are representative substances. This addition would facilitate a better understanding of the formation of polymerization and other effects.
  3. The manuscript lacks an explanation of the mechanism of protein and polyphenol action. I suggest that the authors supplement the relevant content to enhance the completeness of the paper.
  4. The figures and tables in the manuscript are not clear enough, and the differences should be highlighted. Just one figure is not sufficient to express the intended information.

Author Response

The paper entitled ‘Procyanidins and anthocyanins of young and aged Prokupac wines: Evaluation of their reactivity towards salivary proteins’ discussed influence of wine aging on the interaction between phenolic compounds and salivary proteins and emphasize the dominant role of procyanidins in protein binding and the potential synergistic contribution of anthocyanins to mouthfeel perception.The manuscript is quite good as for contents, conclusions are clear and the Ref list is complete and correct.

Thank you very much for your positive and constructive feedback. We truly appreciate your encouraging comments and are glad that you found the manuscript clear, well-structured, and complete.

While I don’t have any particular scientific issues with this paper, there are certain aspects of the manuscript that must be revised.

All suggestions have been accepted and the corresponding corrections have been marked in red.

Below are the specific details:

  1. Figure 1 displays the electrophoretic patterns of salivary proteins prior to and following interaction. It is crucial to note that a well-written manuscript should always be clear and comprehensible. At least, that is what I believe as a reviewer. Therefore, I highly recommend that the authors include a table to visually summarize this figure.

Thank you for this suggestion. To better and more clearly present the results of the electrophoretic analysis, an additional figure (representative electropherograms) and table (results of densitometric analysis) have been included in the manuscript. Please see subsection 3.3, Figure 3, and Table 3 in the revised version of the manuscript.

  1. 2. To improve the readability of the paper, it would be beneficial to add the structural formulas of various anthocyanins, which are representative substances. This addition would facilitate a better understanding of the formation of polymerization and other effects.

Thank you for this suggestion. We agree that including the structural formulas of representative wine phenolic compounds can enhance the readability of the manuscript and improve the interpretation and understanding of the interactions between flavan-3-ols, procyanidins, anthocyanins, and salivary proteins. Accordingly, the structural formulas of representative flavan-3-ols, procyanidins, and anthocyanins identified through untargeted analysis of young and aged Prokupac wines are now presented in Figure 1 (see Figure 1 in the revised manuscript).

  1. The manuscript lacks an explanation of the mechanism of protein and polyphenol action. I suggest that the authors supplement the relevant content to enhance the completeness of the paper.

Thank you for this comment. The mechanism of interaction between wine phenolics and salivary proteins (PRPs) has been further explained and is now supported by a schematic representation. Please see subsection 3.3, subsection 3.4, and Figure 4 in the revised manuscript.

  1. The figures and tables in the manuscript are not clear enough, and the differences should be highlighted. Just one figure is not sufficient to express the intended information.

Thank you for this comment. In accordance with your suggestion, additional explanations (see subsections 3.2, 3.3, and 3.4), figures (Figure 1, Figure 2 – corrected version, Figure 3, and Figure 4), and tables (Table 2, Table S2, Table 3, Table S3) have been included in the revised manuscript. The material and methods section as well as supplementary material has been also extended for necessary information. Please see lines 210-215 and Table S1.

Reviewer 2 Report

Comments and Suggestions for Authors

This article studied in detail the interaction between proanthocyanidins, anthocyanidins and salivary protein in wines with different storage times, as well as the form of the complex. The results show that the types and contents of polyphenolic compounds in wines with different storage times are very different, proanthocyanidins dominate the connection, and the solubility of the complex is significantly affected by the age of the wine. It provides important reference value for understanding the changes in the storage process of wine and the rational application of wine. However, there are some minor issues that need to be raised.
Abstract: It can be more comprehensive, such as the differences in the types and contents of phenols between wines, the binding rate of polyphenols in grape skins and seeds, and the differences in the flavor perception of wine.
Keywords: It can be properly sorted, and it is more logical to put things in front of molecular substances.
Line 134: The amount of phosphate buffer solution added should be supplemented to make the content more complete.
Line 178: The amount of precipitate added should be explained.
Line 288: The differences in the total content of polyphenolic compounds in wines with different storage times and the differences in the content of polyphenols that are easy to bind to salivary protein should be summarized, analyzed and highlight which type of wine proanthocyanidins are more likely to bind to saliva to lay the foundation for the subsequent study of binding rate.
Line 330: There are grammatical errors in this sentence.
Line 383: It should first be shown that there is a clear correspondence between molecular weight and insoluble complexes before this conclusion can be drawn.
Line 429: The differences in taste between wines with different storage times can be analyzed in detail to make the content more complete.

Author Response

This article studied in detail the interaction between proanthocyanidins, anthocyanidins and salivary protein in wines with different storage times, as well as the form of the complex. The results show that the types and contents of polyphenolic compounds in wines with different storage times are very different, proanthocyanidins dominate the connection, and the solubility of the complex is significantly affected by the age of the wine. It provides important reference value for understanding the changes in the storage process of wine and the rational application of wine.

Thank you for your valuable comment. We appreciate your recognition of the study’s focus on the interaction between proanthocyanidins, anthocyanidins, and salivary proteins in wines of different storage times. We are glad that the findings were deemed informative and relevant.

However, there are some minor issues that need to be raised.

All suggestions have been accepted and the corresponding corrections have been marked in red.

Abstract: It can be more comprehensive, such as the differences in the types and contents of phenols between wines, the binding rate of polyphenols in grape skins and seeds, and the differences in the flavor perception of wine.

Thank you very much for this suggestion. The abstract has been revised as per your recommendation.

Keywords: It can be properly sorted, and it is more logical to put things in front of molecular substances.

Thank you for your suggestion. The keywords have been reordered according to your recommendation.

Line 134: The amount of phosphate buffer solution added should be supplemented to make the content more complete.

Thank you very much for this observation. The requested information has been added. Please see line 142 in the revised manuscript.

Line 178: The amount of precipitate added should be explained.

Thank you for this observation. The precipitates refer to the residues remaining after centrifugation of the salivary protein solution/wine mixtures and salivary protein solution/skin or seed mixtures, following the removal of the supernatants. Additional explanation has been added to avoid misunderstanding. See lines 182-184.

Line 288: The differences in the total content of polyphenolic compounds in wines with different storage times and the differences in the content of polyphenols that are easy to bind to salivary protein should be summarized, analyzed and highlight which type of wine proanthocyanidins are more likely to bind to saliva to lay the foundation for the subsequent study of binding rate.

Thank you very much for this observation. An additional subtitle has been introduced to summarize the differences between young and aged wines. Please see Section 3.2.

Line 330: There are grammatical errors in this sentence.

Thank you very much for this observation. The sentence has been corrected. Please see line 383 in the revised manuscript.

Line 383: It should first be shown that there is a clear correspondence between molecular weight and insoluble complexes before this conclusion can be drawn.

Thank you for this observation. The additional explanation has been added in the section 3.3. Please see lines 378-380; 414-452; 467-470; and Figure 3, Table 3, Table S3.

Line 429: The differences in taste between wines with different storage times can be analyzed in detail to make the content more complete.

Thank you for this suggestion. The sensory analysis of the wines has been expanded to provide a more comprehensive overview. Please see lines 236-239, 526-534; 551-561 and Figure 5 and Table S5. The conclusion has been extended to include the sensory results, as reflected in lines 576–579.

Reviewer 3 Report

Comments and Suggestions for Authors

The manuscript is very well written, in a clear and comprehensive manner. The methodology, in particular the experimental plan using electrophoresis experyment, deserves recognition.

The weak point of the methodology is the fact that different materials were studied: young and aged wine. The results would have an even more solid basis if young wine was studied and then the same wine aged after a few years of storage. Nevertheless, the results are general enough that this is not a critical error.

While reviewing the manuscript, I found two editorial errors:
line 246: "(V)" - should be unbold
line 252: "wines. and" - the dot should be removed

Author Response

The manuscript is very well written, in a clear and comprehensive manner. The methodology, in particular the experimental plan using electrophoresis experyment, deserves recognition.

The weak point of the methodology is the fact that different materials were studied: young and aged wine. The results would have an even more solid basis if young wine was studied and then the same wine aged after a few years of storage. Nevertheless, the results are general enough that this is not a critical error.

Thank you for your valuable and encouraging feedback. We appreciate your recognition of the manuscript’s clarity and the experimental design, as well as your constructive comment regarding the sample selection, which we will consider for future research.

While reviewing the manuscript, I found two editorial errors:

line 246: "(V)" - should be unbold

Thank you very much. The correction has been made. Please see line 260 in the revised manuscript. The change has been marked in red.

line 252: "wines. and" - the dot should be removed

line 246: "(V)" - should be unbold

Thank you very much. The correction has been made. Please see line 266 in the revised manuscript. The change has been marked in red.